# Harmonic Loss Trains Interpretable AI Models

**David D. Baek**[*]                                                   *dbaek@mit.edu*
*Massachusetts Institute of Technology*

**Ziming Liu**[*]                                                     *zmliu@mit.edu*
*Massachusetts Institute of Technology*

**Riya Tyagi**                                                       *riyaty@mit.edu*
*Massachusetts Institute of Technology*

**Max Tegmark**                                                    *tegmark@mit.edu*
*Massachusetts Institute of Technology*

**Reviewed on OpenReview:** *https://openreview.net/forum?id=ZpSZ7pNoCs*

## Abstract

In this paper, we introduce **harmonic loss** as an alternative supervisory signal for training neural networks and large language models (LLMs). Harmonic loss differs from standard cross-entropy loss by (a) replacing the usual SoftMax normalization with a scale-invariant HarMax function and (b) computing logits via Euclidean distance rather than a dot product. Harmonic loss enables improved interpretability and faster convergence, owing to its scale invariance and finite convergence point by design, which can be interpreted as a class center. We first validate the performance of harmonic models across algorithmic, vision, and language datasets. Through extensive experiments, we demonstrate that models trained with harmonic loss perform better than standard models by: (a) enhancing interpretability (i.e. geometry of representations), (b) requiring less data for generalization, and (c) reducing grokking. Moreover, we compare a GPT-2 model trained with harmonic loss to the standard GPT-2, illustrating that the harmonic model develops more interpretable representations. We hope our work will inspire future research exploring various methods to improve the geometry of representations, paving the way toward building more interpretable AI models.

## 1 Introduction

As machine learning models become powerful, it has become increasingly important to thoroughly understand the behavior of neural networks. One particularly intriguing characteristic of neural networks is their ability to generalize: Empirical evidence shows that neural networks can perform well on unseen data not explicitly encountered during training (Novak et al., 2018). This remarkable ability stems from the networks' capacity to learn generalizable representations and algorithms through training. However, current models face three key challenges when it comes to generalization:

**(1) Lack of interpretability:** Neural networks often lack interpretability, which is a critical issue in high-stakes applications like healthcare, finance, and autonomous systems. While multiple research efforts have advanced our insight into inner workings of LLMs (Bereska and Gavves, 2024), we are still far from fully explaining their outputs. Ultimately, it is crucial to design systems that are interpretable by design. Otherwise, it is challenging to diagnose errors, ensure fairness, or build trust in a model's decisions.

**(2) Low data efficiency:** Generalization often requires vast and diverse training data. This raises a critical question: can models generalize effectively with less data? This issue is especially relevant in domains where

---

[*]Equal contribution.

data availability is scarce, such as rare disease diagnosis or specialized scientific fields. Previous approaches for improving neural network generalization include efficient data sampling (Li et al., 2024a) and modifications to the training procedure to accelerate training (Wang et al., 2024). However, these methods focus on optimizing existing training procedures rather than addressing the core issues in model design.

**(3) Delayed generalization (grokking):** Models sometimes experience a phenomenon known as "grokking," (Power et al., 2022; Liu et al., 2022a) where there is a noticeable delay between the convergence of the training loss and the convergence of the test loss. This gap is problematic because: (i) it complicates determining the optimal point to stop training in order to achieve generalization, and (ii) it necessitates extended computation time and resources to continue training until grokking occurs.

A contributing factor may be how probabilities are induced from logits. In particular, Softmax's temperature-sensitive normalization interacts with scale and margin in ways that can complicate optimization and geometry. Thus, we propose **harmonic loss** as an alternative. Harmonic loss has two desirable mathematical properties that enable faster convergence and improved interpretability: (1) scale invariance, and (2) a finite convergence point, which can be interpreted as a class center. We also present a range of empirical evidences that harmonic loss can reduce grokking, match or improve data efficiency, and produce more structured representations relative to standard cross-entropy.

Furthermore, we compare a GPT-2 model trained with harmonic loss to the standard GPT-2 and show that the harmonic model develops more interpretable representations. While its scalability to even larger models remains to be evaluated in future works, we believe that the scale-invariant formulation of the harmonic loss is crucial for enhancing representation geometry. We hope our work will inspire more future works on developing training methods that yield interpretable representations.

The remainder of this paper is organized as follows: We first review the relevant literature in Section 2. Section 3 introduces the principles underlying harmonic loss and explains why it is preferable to cross-entropy loss in terms of generalization and interpretability. Section 4 details a comprehensive set of experiments on algorithmic datasets, illustrating that models trained with harmonic loss have numerous desirable properties that are absent in standard models. In Section 5, we demonstrate the performance of harmonic models on the vision task of MNIST digit classification. In Section 6, we extend our analysis to large models, illustrating that the advantages of harmonic loss also hold at scale. We present ablation experiments in Section 7. We conclude the paper in Section 8.

## 2  Related Works

**Representations and Mechanistic Interpretability:** Numerous studies have shown that LLMs can form conceptual representations across spatial (Gurnee and Tegmark, 2023), temporal (Li et al., 2021), and color domains (Abdou et al., 2021). The structure of such representations includes one-dimensional concepts (Gurnee and Tegmark, 2023; Marks and Tegmark, 2023; Heinzerling and Inui, 2024; Park et al., 2024a), as well as multi-dimensional representations such as lattices (Michaud et al., 2024; Li et al., 2024b; Park et al., 2024b) and circles (Liu et al., 2022a; Engels et al., 2024). While the structure of these representations correlates with certain geometric patterns, significant unexplained variance frequently remains, necessitating efforts to improve the interpretability of neural network representations.

**Loss Functions:** Previous research has shown that loss functions can influence how a model learns to represent data, affecting its abilities in unique ways (Li et al., 2024c; Bosco et al., 2024; Sudre et al., 2017; Demir et al., 2023; Salehi et al., 2017; Bommidi et al., 2023; Seber, 2024). We refer readers to (Alshammari et al., 2025) and (Wang et al., 2022) for a comprehensive survey of different loss functions used in machine learning. Our harmonic loss offers an alternative supervisory signal in standard supervised learning by (a) replacing the usual SoftMax normalization with a scale-invariant HarMax function and (b) computing logits via Euclidean distance rather than a dot product. While it bears resemblance to contrastive loss, which also encourages maximal separation between different classes by using Euclidean distance as a metric, contrastive learning methods are not inherently supervised: they typically append a cross-entropy layer to generate logits, thus reintroducing SoftMax (and its drawbacks). We also show in Section 7 that using Euclidean distance alone is insufficient to fully replicate harmonic loss's capabilities. Furthermore, directly leveraging Euclidean

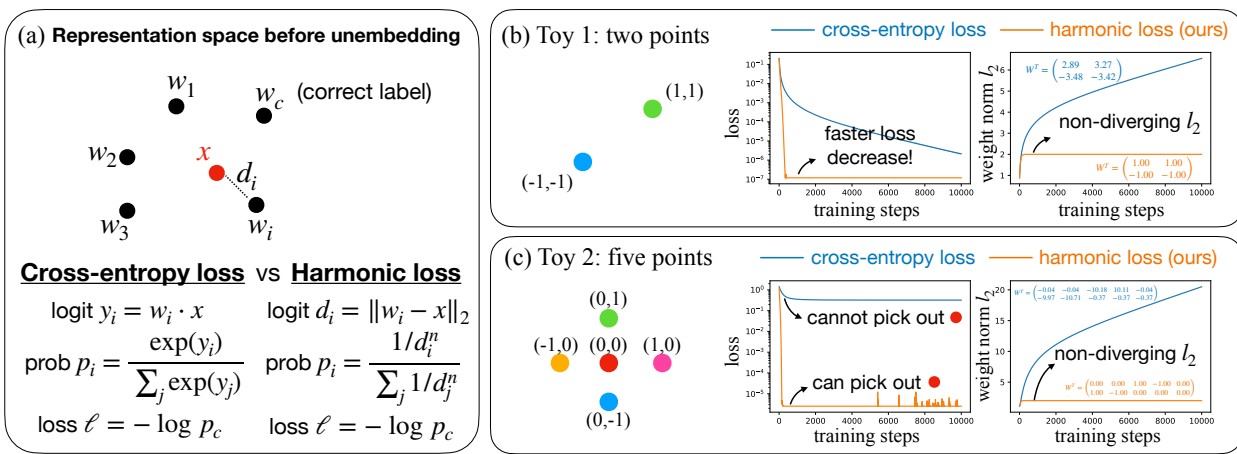

Figure 1: Cross-entropy loss versus harmonic loss (ours). (a) Definitions. Cross-entropy loss leverages the inner product as the similarity metric, whereas the harmonic loss uses Euclidean distance. (b) Toy case 1 with two points (classes). Both the loss and $l_2$ weight norm converge faster for the harmonic loss. (c) Toy case 2 with five points (classes). Harmonic loss can pick out the red point in the middle. By contrast, the cross-entropy loss cannot, since the red point is not linearly separable from other points. Weight matrices are also more interpretable with harmonic loss than with cross-entropy loss.

distance-based supervised learning has been relatively underexplored in language modeling outside of simple tasks like sentence sentiment classification (Xu et al., 2023). We present a more comprehensive comparison of harmonic loss with other loss functions in Appendix D. We also note that after completing this project, we became aware of a prior work that explored an idea similar to ours, which the authors term inverse distance weighting attention (McCarter, 2023), although their empirical evaluation is limited in scope and does not comprehensively examine the benefits of employing inverse distance weighting.

**Prototype Learning:** Our harmonic loss is closely related to the prototype learning literature, which aims to learn vectors that represent each class region in the feature space. A family of algorithms designed for such prototype-based classification is known as Learning Vector Quantization (LVQ) (Kohonen, 1990; Sato and Yamada, 1995; Nova and Estévez, 2014). Recently, prototype-based methods have been extended to modern deep learning settings (Sharma et al., 2023; Guerriero et al., 2018; Chen et al., 2019; Deng et al., 2009), showing particular promise in improving interpretability in image classification tasks. In this paper, we take a modest step toward exploring loss functions that are suitable for learning prototypes across a wide range of domains, including algorithmic, vision, and language modeling tasks.

## 3  Harmonic Loss

We first review cross-entropy loss and present the harmonic loss, visualized in Figure 1 (a). Denote the unembedding matrix as $\boldsymbol{W} \in \mathbb{R}^{N \times V}$ ($N$ is the embedding dimension, $V$ is the vocabulary size), and the penultimate representation (the representation prior to the unembedding matrix) as $\boldsymbol{x} \in \mathbb{R}^N$.

**Cross-entropy loss:** Logits $\boldsymbol{y}$ are defined as the matrix-vector multiplication, i.e., $\boldsymbol{y} = \boldsymbol{W}^T \boldsymbol{x} \in \mathbb{R}^V$ (ignoring biases), or $y_i = \boldsymbol{w}_i \cdot \boldsymbol{x}$, where $\boldsymbol{w}_i$ is the $i^{\text{th}}$ column of $\boldsymbol{W}$. Probability $\boldsymbol{p}$ can be obtained by applying SoftMax to $\boldsymbol{y}$, i.e.,

$$p_i = \text{SoftMax}(\boldsymbol{y})_i \equiv \frac{\exp(y_i)}{\sum_j \exp(y_j)}. \tag{1}$$

Suppose the real class label is $c$, then loss $\ell = -\log p_c$. For notational simplicity, we call a linear layer combined with the cross-entropy loss a *cross-entropy layer*.

**Harmonic loss:** The *harmonic logit* $\boldsymbol{d}$ is the $l_2$ distance between $\boldsymbol{w}_i$ and $\boldsymbol{x}$, i.e., $d_i = ||\boldsymbol{w}_i - \boldsymbol{x}||_2$. We interpret $\boldsymbol{w}_i$ as keys and $\boldsymbol{x}$ as a query, so smaller $d_i$ means a higher probability of $p_i$. We define *harmonic max (HarMax)* as

$$p_i = \text{HarMax}(\boldsymbol{d})_i \equiv \frac{1/d_i^n}{\sum_j 1/d_j^n}, \tag{2}$$

where $n$ (*harmonic exponent*) is a hyperparameter that controls the heavy-tailedness of the probability distribution. If the true class label is $c$, then loss $\ell = -\log p_c$. For notational simplicity, we call a layer combined with the harmonic loss the *harmonic layer*. Since the last step of both losses is the same ($\ell = -\log p$), comparing their values is meaningful. They only differ in the ways of computing probabilities from representations [1].

A reasonable choice of $n$ is $n \sim \min(\sqrt{D_1}, \sqrt{D_2})$, where $D_1$ is the embedding dimension, and $D_2$ represents the intrinsic dimensionality of the underlying data. In LLMs, $D_2$ could be approximated as $D_2 \approx d_{\text{embed}}$, where $d_{\text{embed}}$ is the embedding dimension. This approximation arises from considering an embedding initialized from a $D$-dimensional Gaussian distribution. The squared distance between two points, normalized by the number of dimensions $D$, is on the order of $1 \pm O(1/\sqrt{D})$. To ensure that the harmonic distance $\left[1 \pm O(1/\sqrt{D})\right]^n$ remains constant as we scale $D$, we require $n \sim \sqrt{D}$, since $\lim_{x \to \infty}(1 + x^{-1})^x = e$. We also show the empirical impact of the exponent on the learned representations in Appendix E.

**Toy cases:** To provide intuition about what advantages the harmonic loss has over the cross-entropy loss, we consider two toy cases in 2D, as shown in Figure 1 (b)(c). In each toy case, we train the cross-entropy layer and the harmonic layer with the Adam optimizer. **Toy case 1**: $\boldsymbol{x}_1 = (1, 1)$ and $\boldsymbol{x}_2 = (-1, -1)$ belong to two different classes. The harmonic layer produces a faster loss decrease, because the harmonic loss only requires $d_i \to 0$ (converging point is finite) to get $p_i \to 1$. By contrast, cross-entropy loss requires $y_i \to \infty$ (converging point is infinite) to get $p_i \to 1$. The harmonic loss already produces a $l_2$ weight norm that plateaus to a constant, while the cross-entropy loss leads to increasing $l_2$, diverging towards infinity. **Toy case 2**: There are 5 points in 2D, each of which belong to a different class. In particular, the red point $(0, 0)$ is surrounded by the other four points, i.e., cannot be linearly separated. The cross-entropy layer indeed cannot perform well on this task, manifested by a high loss plateau. By contrast, the harmonic layer can drive the loss down to machine precision. Similar to case 1, the harmonic layer has a plateauing $l_2$ while the cross-entropy layer has an ever-growing $l_2$. We also observe that the weights of the harmonic layer correspond to $\boldsymbol{x}$, which is more interpretable than the weights of the cross-entropy layer.

**Benefits of harmonic loss:** From these two toy cases, we understand the advantages of harmonic loss: (1) *nonlinear separability*: in case 2, the red dot can be classified correctly even though it is not linearly separable. (2) *fast convergence*: The fact that the converging point is finite leads both to faster loss decay, and plateauing (non-diverging) $l_2$. (3) *scale invariance*: Harmonic loss is scale-invariant, i.e., $d_i \to \alpha d_i$ leaves $p_i$ (hence loss) invariant, whereas $y_i \to \alpha y_i$ would produce a different cross-entropy loss. (4) *interpretability*: the weight vectors correspond to class centers. We present the formal proof of these properties in Appendix F.

**Notes on interpretability:** Measuring interpretability is inherently challenging in the absence of ground-truth representations. Hence, we propose two principled indicators of interpretability throughout the paper: (1) *Compression*: Sparse, low-dimensional representations enhance interpretability by concentrating semantics. We measure this via cumulative explained variance in PCA projections. (2) *Geometry*: In general models, we hypothesize that parallelogram-like units with multiple one-dimensional semantic directions enable compositional reasoning; This enables vector arithmetic such as *man – woman = king – queen*, and supports faithful feature attribution. We measure this via parallelogram loss in Section 6.

## 4 Algorithmic Experiments

Algorithmic tasks are good benchmarks for interpretability since they are well-defined mathematically. However, training neural networks on these tasks is non-trivial due to grokking (delayed generalization) (Power

---

[1]Note that when we say "cross-entropy loss," we do not only refer to $\ell = -\log p$, but rather refer to the whole pipeline including penultimate representation, logit, probability, and loss.

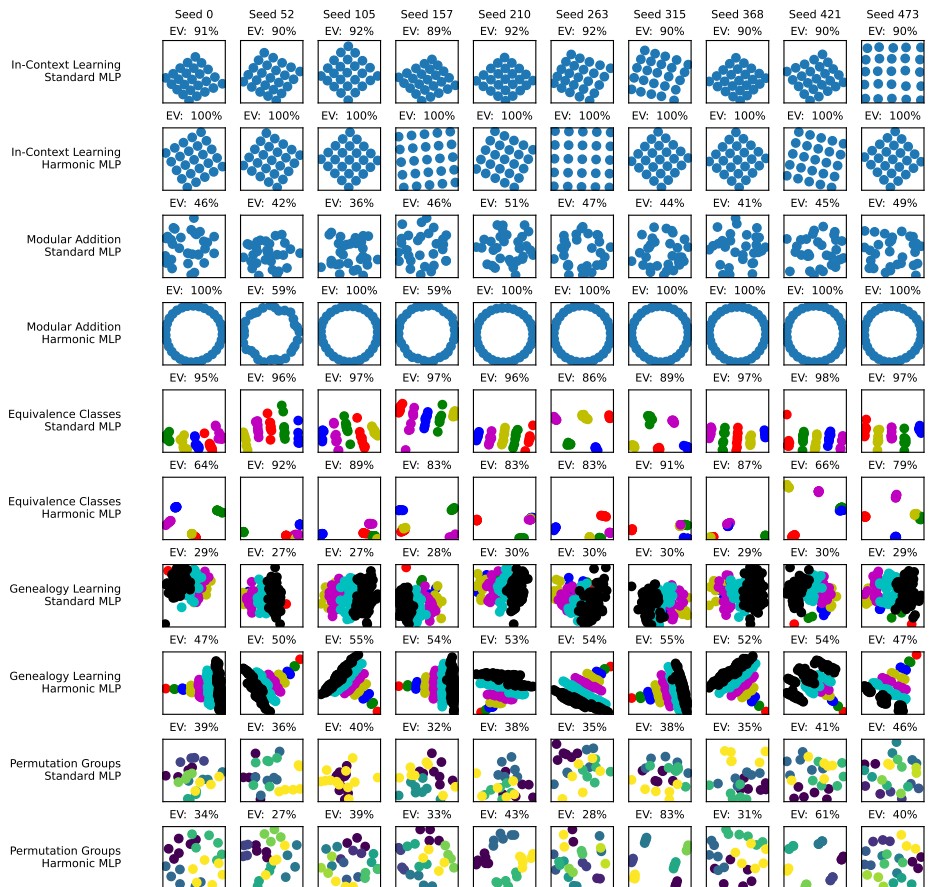

Figure 2: Visualization of the top two principal components of the embeddings in synthetic experiments. The title of each subplot shows the explained variance by the first two principal components. Each row corresponds to a pair of a dataset and a model, while each column represents the embeddings from different training runs with varying seeds. Groups of consecutive two rows belong to the same dataset, with models arranged in the order: {Standard MLP, Harmonic MLP}. The datasets are ordered as follows: {In-Context Learning, Genealogy Learning, Equivalence Classes, Modular Addition, and Permutation Groups}. X and Y axis spans are equal.

et al., 2022) and the existence of multiple algorithms (Zhong et al., 2024), etc. We will show that harmonic models learn better representations, are more data-efficient, and exhibit less grokking.

## 4.1 Models and Datasets

**Models:** We compare four models:

1. **Standard MLP**: Tokens are embedded into 16-dimensional embeddings, which are then concatenated and used as the input. The model consists of two hidden layers with widths of 100 and 16, respectively. The SiLU activation function is used.

2. **Standard Transformer**: Tokens are embedded into a 16-dimensional embedding, with a learnable positional embedding added. The input passes through two transformer decoder layers, each comprising two attention heads and an MLP with a hidden dimension of 64.

3. **Harmonic MLP**: Standard MLP with an harmonic unembedding layer of exponent $n = 1$.

4. **Harmonic Transformer**: Standard Transformer with an harmonic unembedding layer of exponent $n = 1$.

We trained the MLP models for 7000 epochs and the transformers for 10000 epochs. For all four models, we used the AdamW optimizer with a learning rate of $2 \times 10^{-3}$, a weight decay of $10^{-2}$, and an $L_2$ regularization on the embeddings with strength 0.01.

**Datasets:** We trained the four models above using the following five datasets, and analyzed their performance as well as the resulting representations:

1. **In-Context Learning**: In a $5 \times 5$ integer lattice, given three points on the lattice, the model is trained to predict the fourth point that would form a parallelogram with the others. This task exemplifies in-context reasoning in LLMs, mirroring the classic *man:woman::king:queen* analogy by requiring the model to complete the relational pattern such as 'man is to woman as king is to queen' based on the given context.

2. **Modular Addition**: Given two integers $x, y$, the model is trained to predict $(x + y)$ mod 31.

3. **Equivalence Classes**: Given two integers $0 \leq x, y < 40$, the model is trained to predict if $x \equiv y$ mod 5.

4. **Genealogy Learning**: In a complete binary tree with 127 nodes, given a subject and a relation, the model is trained to predict the corresponding object. The relation can be one of the following: parent, grandparent, or sibling.

5. **Permutation Composition**: Given two permutations $x$ and $y$ in $S_4$, the model is trained to predict $x \circ y$. On this dataset, we trained standard and harmonic transformers with an $L_2$ regularization of 0.005, as we found this configuration led to more complete training.

### 4.2 Representation Faithfulness

Figure 2 shows the plot of the top two principal components of the models' embeddings for MLP tasks. We show the complete embedding visualization for all tasks in Appendix A. Overall, harmonic loss representations are cleaner and more organized than their cross-entropy counterparts. We found near-perfect circle representations for the modular addition task, a clear tower-like structure for tree learning, and neat clusters for permutation composition. We examine the representations task by task:

1. **In-context Learning**: Standard models' representations are either imperfect lattices or exhibit unexplained variance in higher dimensions, whereas harmonic models almost always perfectly (100%) recover the underlying 2D lattice structure across different random seeds.

2. **Modular Addition**: Harmonic MLPs consistently recover a perfect 2D circular representation in almost all runs, whereas tstandard MLPs often fail to do so. Harmonic transformer has a similar success rate to the standard transformer in constructing circles, but the explained variance captured by the first two principal components is generally much higher, indicating that harmonic models discover more compact representations with fewer uninterpretable components.

3. **Equivalence Classes**: While both standard and harmonic models are able to identify the underlying groups, standard models' representation tends to be more "elongated", or not *completely* grouped, compared to its harmonic counterpart. This could be attributed to the fact that cross-entropy loss does not have an incentive to reduce irrelevant variations to zero.

4. **Genealogy Learning**: Only Harmonic MLP recovers the underlying tree representation.

5. **Permutation Composition**: Harmonic MLP generally produces better-separated clusters. A particularly clean representation that appears multiple times contains 6 clusters of 4 permutations, where each cluster is a coset of the subgroup $\langle e, (12)(34), (13)(24), (14)(23) \rangle$ or one of its conjugates. In the harmonic transformer, permutations commonly organize into 4 clusters that are cosets of $\langle e, (13), (14), (34), (134), (143) \rangle$ or one of its conjugates, subgroups isomorphic to $S_3$ (one element, in this case 2, never permutes).

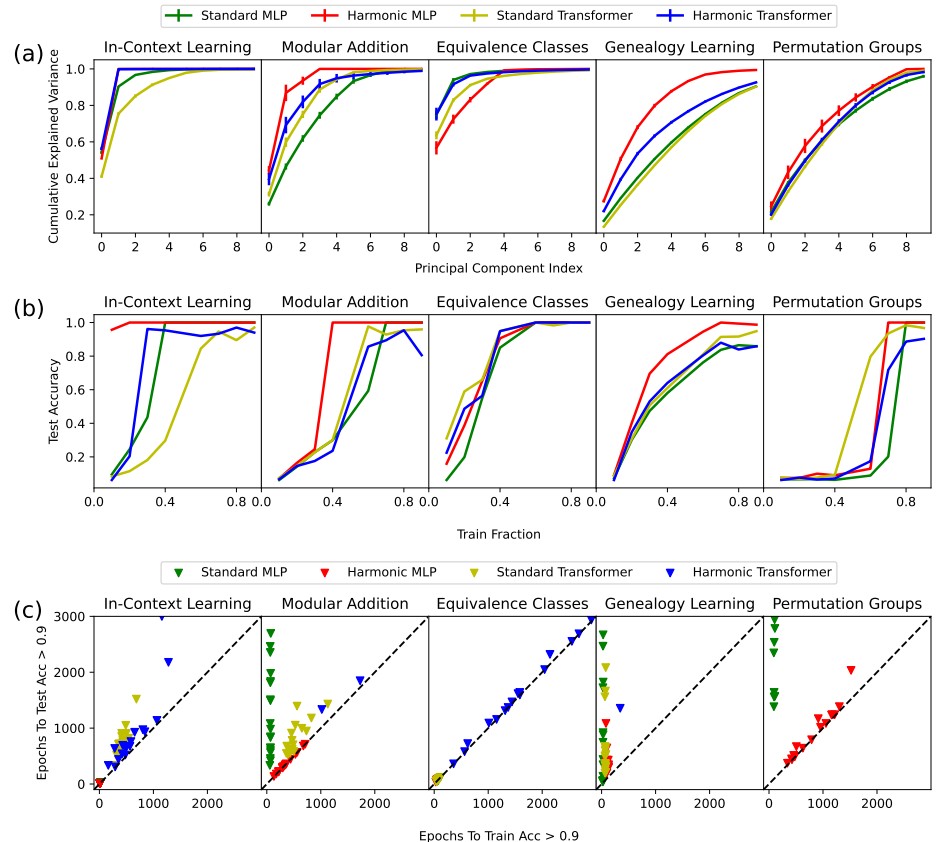

Figure 3: (a) Cumulative explained variance vs. principal components (mean over 20 seeds). Harmonic representations are more compact than standard counterparts. (b) Test accuracy as a function of Train Fraction (fixed seed). Harmonic models generalize faster with less data than standard counterparts. (c) Epochs to Test Acc > 0.9 vs. Epochs to Train Acc > 0.9 for 20 consecutive epochs, for 20 different random seeds. $y = x$ line represents no grokking, where train and test accuracy improve simultaneously. Points closer to the y-axis indicate a greater degree of grokking.

Figure 3(a) further demonstrates that harmonic representations tend to be more compact than standard models, with fewer uninterpretable components. In particular, harmonic models trained for in-context learning achieve 100% explained variance using only the first two principal components.

### 4.3 Data Efficiency in Training

Figure 3(b) shows the test accuracy as a function of train data fraction for our synthetic experiments, indicating how much data is necessary in order for the model to be generalizable. We observe that harmonic models require comparable or much less amount of data to generalize, compared to their cross-entropy counterparts. Such improvement is especially notable for in-context learning, where harmonic models generalize nearly immediately.

### 4.4 Reduced Grokking

Grokking refers to the phenomenon of delayed generalization (Power et al., 2022): for example, it takes $10^3$ steps to reach perfect accuracy on the training data, but it takes $10^5$ steps to generalize to the test data. Grokking is a pathological phenomenon that we want to avoid (Liu et al., 2022b). We find that harmonic loss overall reduces grokking, as seen in Figure 3(c). Points on the $y = x$ line represent models which trained without grokking, with train and test accuracy improving together. This improvement is particularly evident

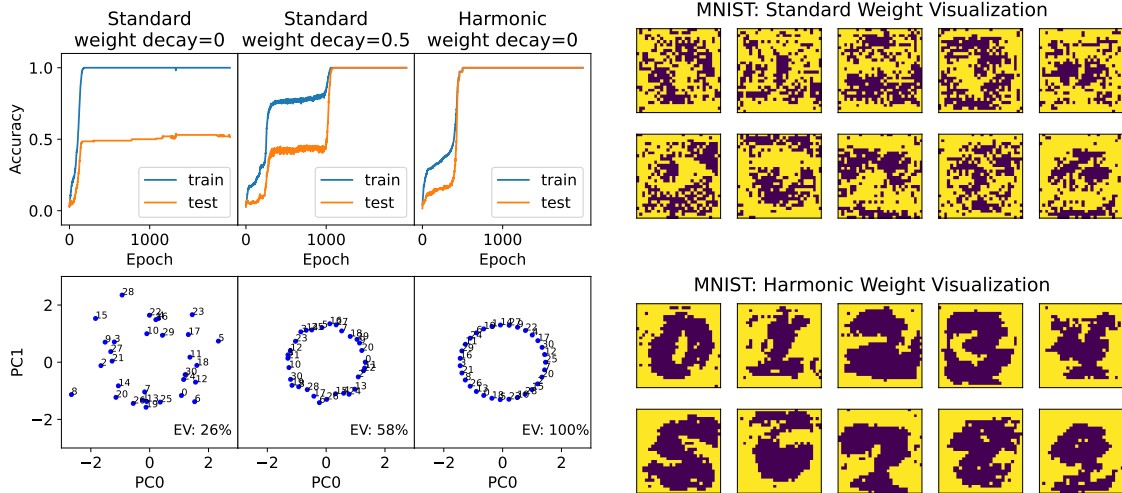

Figure 4: **Left**: Case study on modular addition. Standard MLP trained for modular addition without weight decay often fails to generalize. Generalization is only achieved with the addition of strong weight decay; however, (a) significant grokking occurs, and (b) while the first two principal components form an approximate circle, they explain far less than the total variance. In contrast, the harmonic model trained for modular addition generalizes quickly without grokking. Moreover, the embedding forms a perfect 2D circle. EV in the plot represents the explained variance by the first two principal components of the embedding. **Right**: Visualization of model weights trained for MNIST. Yellow cells show values less than 0.01. Both models achieved $\approx 92.5\%$ test accuracy.

in learning modular addition and permutation composition: while the standard MLP exhibits severe grokking, most data points for the harmonic MLP lie much closer to the $y = x$ line.

## 4.5 Case Study: Modular Addition

In this section, we study modular addition as a case study and analyze why the harmonic MLP encourages more interpretable representations and better generalization compared to the standard MLP. The standard MLP trained for modular addition without weight decay often fails to generalize, as shown in Figure 4. Generalization is only achieved with the addition of strong weight decay; however, (a) significant grokking occurs, as depicted in Figure 4, and (b) while the first two principal components form an approximate circle, they explain far less than the total variance, leaving significant unexplained variance. In contrast, the harmonic model trained for modular addition generalizes quickly without grokking. Furthermore, the embedding forms a perfect circle, as shown in Figure 4.

The better formation of a circle and improved generalization in harmonic MLP can be attributed to the properties of harmonic loss, as explained in Section 3. To drive the probability to 1, the standard cross-entropy loss requires driving the representation to infinity—*i.e.*, making the logit infinite. In contrast, harmonic loss achieves this by driving the harmonic logit to zero, which is easily accomplished by learning $\boldsymbol{w}_i = \boldsymbol{x}$ in Equation 2. The existence of such a finite converging point may be responsible for the (a) faster convergence, (b) better generalization, and (c) circular geometry observed in embeddings.

## 5 MNIST Experiments

For vision tasks, convolutional neural networks are shown to be (at least somewhat) interpretable by demonstrating "edge detectors", "wheel detectors", etc. (Olah et al., 2020). In this section, we show that the harmonic loss can lead to a more interpretable network for the MNIST dataset when it comes to training fully connected networks. As a proof of concept, we compare one-layer neural networks trained using cross-entropy

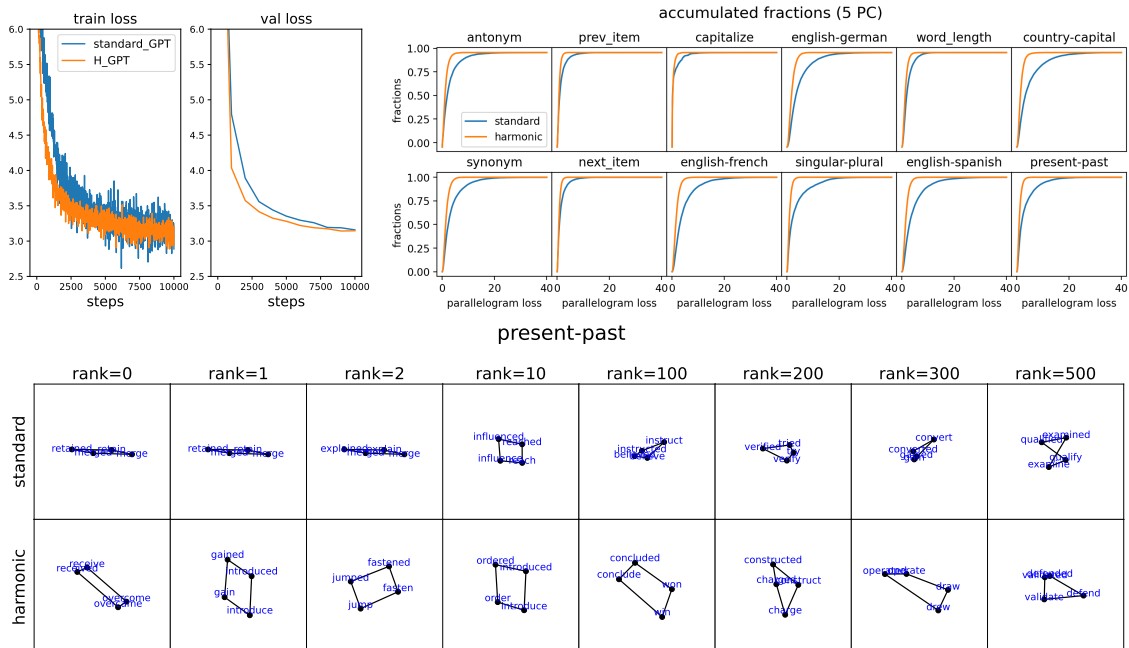

Figure 5: **GPT2 experiments**: (Top left) loss curves. Harmonic GPT achieves a slightly lower loss compared to standard GPT. (Top right) cumulative distribution function with respect to parallelogram loss, for twelve function-vector tasks. Harmonic GPT consistently shows lower parallelogram losses (i.e., better parallelograms). (Bottom) Parallelograms (1st and 2nd principal component) with quality ranked in descending order from left to right. Harmonic GPT tends to produce parallelograms that are more 'rectangular', while standard GPT produces flat 'parallelograms'.

loss and harmonic loss. The input images are first flattened and passed through a $784 \times 10$ linear layer to obtain the logits. The models were trained with a batch size of 64, a learning rate of 0.001, and for 10 epochs, achieving a 92.50% test accuracy for cross-entropy loss and 92.49% test accuracy for harmonic loss.

Figure 4 shows that the harmonic model's weights are more interpretable than those of the standard model. Consistent with its core principle, the harmonic model's weights almost perfectly align with class centers (images of each number). They also assign near-zero values to peripheral pixels, unlike the model trained with cross-entropy loss, which lacks an incentive to reduce irrelevant background weights to exactly zero.

## 6 GPT2 Experiments

Many mechanistic interpretability works have been dedicated to understanding large language models. For example, probing and attribution methods are good post hoc analysis tools. Despite their (partial) success, these tools are not creating interpretable models in the first place but are trying to find needles in the haystack. We argue that it would be nicer if we could pre-train the language models to be more interpretable. By using harmonic loss in training, we can produce a language model that can "grow" crystal-like representations, while having comparable performance with a standard one (trained with the cross-entropy loss).

We pre-train a GPT-2 small model (128M, based on NanoGPT) on OpenWebText. The embedding matrix and the unembedding matrix are tied (share the same weights). We use 8 V100 GPUs, choose block size 1024, batch size 480 blocks. We use the Adam Optimizer with $\beta_1 = 0.9$, $\beta_2 = 0.95$. For the harmonic loss, we choose $n = \sqrt{768} \approx 28$, following the discussion on harmonic exponent in Section 3. For standard (harmonic) GPT, we use a linear warmup learning rate schedule for 2k (1k) steps to maximum learning rate $6 \times 10^{-4}$ ($6 \times 10^{-3}$), and a cosine decay schedule from 2k to 10k, ending at lr $3 \times 10^{-5}$ ($3 \times 10^{-4}$). As shown in Figure 5 top left, Harmonic GPT shows faster converging initially (partially due to larger learning rates), and

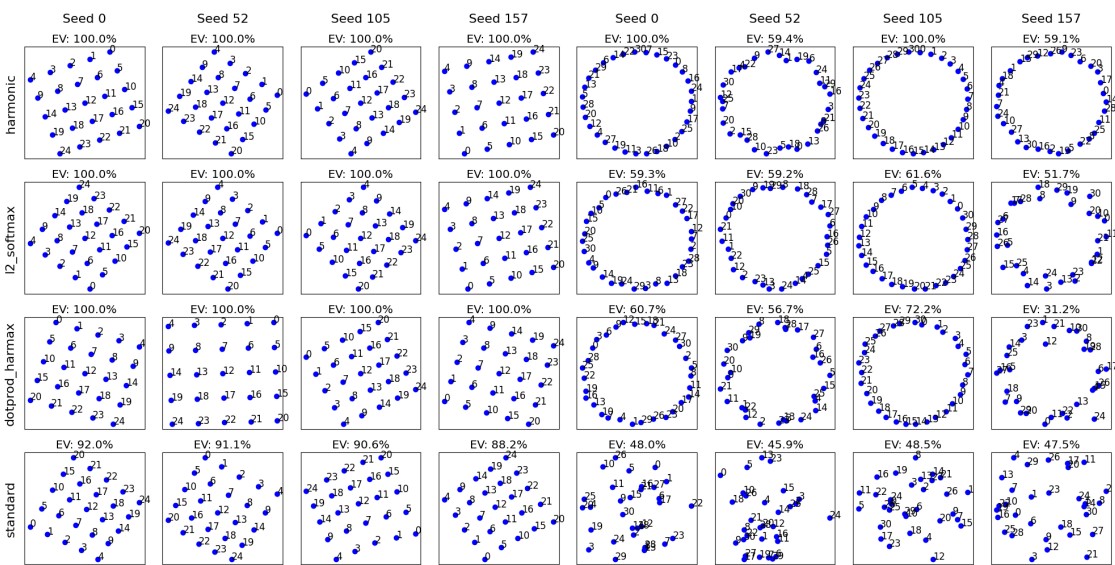

Figure 6: Learned embeddings on the lattice and modular addition tasks. Each pane shows the 5×5 class embeddings after training (numbers denote class IDs). Columns vary random seeds; the four left columns are for in-context learning, and the four right columns are for modular addition task. Rows correspond to loss functions: **(top)** full harmonic loss ($\ell_2$ logits + HarMax), **(2nd)** $\ell_2$ logits + SoftMax, **(3rd)** dot-product logits + HarMax, **(bottom)** standard cross-entropy layer. Here, we see that only $\ell_2$ distance paired with HarMax successfully recovers both the lattice and circular structure.

converges to similar performance in the end (at 10k steps). The final validation losses are 3.159 (standard) and 3.146 (harmonic). From training loss curves, harmonic GPT also seems to have smaller fluctuations. This suggests the effectiveness of the harmonic loss on real-world models.

To testify the interpretability of the learned embeddings, we take twelve function-vector tasks from (Todd et al., 2023). Each dataset contains many input-output pairs that have a certain relation. For example, the "present-past" dataset contains pairs like: jump-jumped, fasten-fastened, win-won, etc. To construct parallelograms, we can draw two different pairs from the dataset, obtaining quadruples like (jump, jumped, fasten, fastened) which are expected to form parallelograms. Each word is tokenized into tokens; if multiple tokens are obtained, we use the last token. We project token embeddings onto the first two principal components. The quadruple $(i, j, m, n)$ has 2D PC embeddings $(\boldsymbol{E}_i, \boldsymbol{E}_j, \boldsymbol{E}_m, \boldsymbol{E}_n)$; we define the parallelogram loss $l_{\mathrm{para}}$ to be

$$l_{\mathrm{para}} = \|\boldsymbol{E}_i + \boldsymbol{E}_n - \boldsymbol{E}_j - \boldsymbol{E}_m\|/\sigma, \tag{3}$$

where $\sigma = \sqrt{\frac{1}{V}\sum_{k=1}^{V}\|\boldsymbol{E}_k\|^2}$ is a scale factor that normalizes the loss ($\boldsymbol{E}_k \to a\boldsymbol{E}_k$ leaves $l_{\mathrm{para}}$ invariant). We obtain 10000 quadruples, measuring the parallelogram qualities by computing their parallelogram losses. We plot their cumulative distribution function in Figure 5 in the top right: for every task, the harmonic GPT produces lower parallelogram loss (better parallelograms) than standard GPT. We show the parallelograms obtained in the present-past task in Figure 5 bottom. The parallelograms are ranked with quality in descending order from left to right. The harmonic GPT tends to produce visually appealing parallelograms that are more 'rectangular', while standard GPT produces flat 'parallelograms'. Discussion about internal representations is included in Appendix C.

## 7   Ablation Experiments

Harmonic loss makes two major modifications to the standard cross-entropy loss: *(i)* compute logits via $\ell_2$ distances, and *(ii)* use HarMax function as shown in Eq. equation 2. To tease apart their individual contributions, we perform a set of targeted ablations in which one component is replaced at a time while the

remainder of the training pipeline is left unchanged. Specifically, we train MLP models on the in-context learning and modular addition tasks with the ablated loss functions.

Results are shown in Figure 6. In in-context learning tasks, we observe that including either HarMax or $\ell_2$ logits alone is sufficient to replicate the full performance of Harmonic Loss. However, for modular addition tasks, both HarMax and $\ell_2$ logits are essential to achieve the full performance. While incorporating only one component enhances the quality of the circular representation, the explained variance remains significantly below 100%. Overall, both HarMax and $\ell_2$ logits play critical roles in improving representations.

## 8 Conclusions

In this paper, we introduced harmonic loss as an alternative to the standard cross-entropy loss for training neural networks and large language models (LLMs). Across the evaluated tasks, we found that models trained with harmonic loss perform better than standard models by: (a) reducing grokking, (b) requiring less data for generalization, and (c) improving interpretability. We also compared a GPT-2 model trained with harmonic loss to the standard GPT-2, illustrating that the harmonic loss-trained model develops more interpretable representations. Further study is needed to explore the scalability and applicability of our findings to even larger models.

**Limitations:** As discussed earlier in the manuscript, measuring interpetability is inherently challenging in the absence of ground-truth representations. For example, although harmonic loss slightly outperforms cross-entropy loss on the ImageNet benchmark (see Appendix G), it is difficult to define ground-truth representations of images, beyond the qualitative observation that semantically similar images tend to cluster in the representation space. This makes it hard to quantify progress in interpretability. Nevertheless, we conjecture that the scale-invariant formulation improves the parallelogram-like geometry of model representations as demonstrated in this paper, which may facilitate identifying semantic feature directions. Further investigation is required to refine our understanding of the extent to which representation geometry affects interpretability. Lastly, we note that harmonic loss might not be optimal in all instances. For instance, we expect cross-entropy loss to be better for tasks requiring linear separability, since they are more suitable for finding the linear separation boundary compared to distance-based losses.

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

## A    Full Representation Visualization

Figure 7 shows the visualization of representations for all models and datasets.

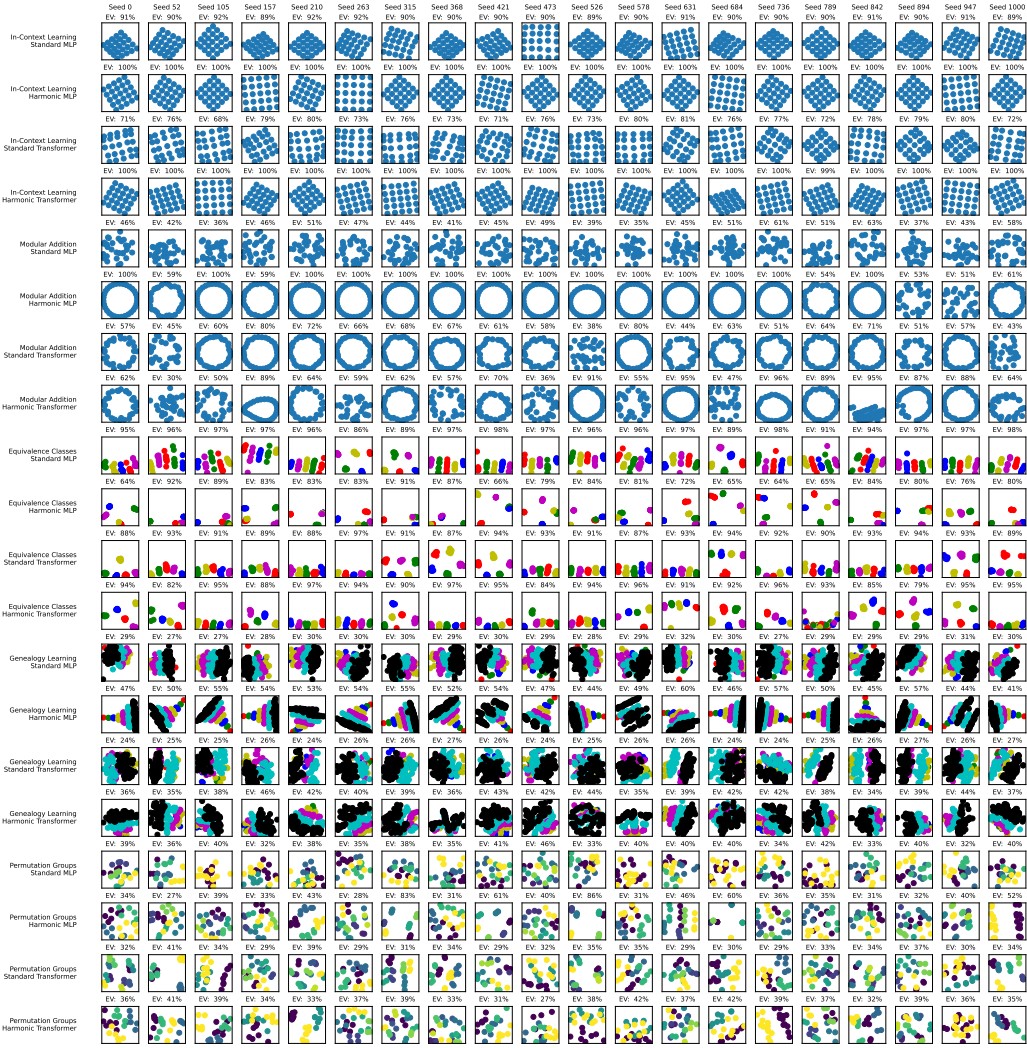

Figure 7: Visualization of the top two principal components of the embeddings in synthetic experiments. The title of each subplot shows the explained variance by the first two principal components. Each row corresponds to a pair of a dataset and a model, while each column represents the embeddings from different training runs with varying seeds. Groups of four rows belong to the same dataset, with models arranged in the order: {Standard MLP, Harmonic MLP, Standard Transformer, Harmonic Transformer}. The datasets are ordered as follows: {In-Context Learning, Genealogy Learning, Equivalence Classes, Modular Addition, and Permutation Groups}.

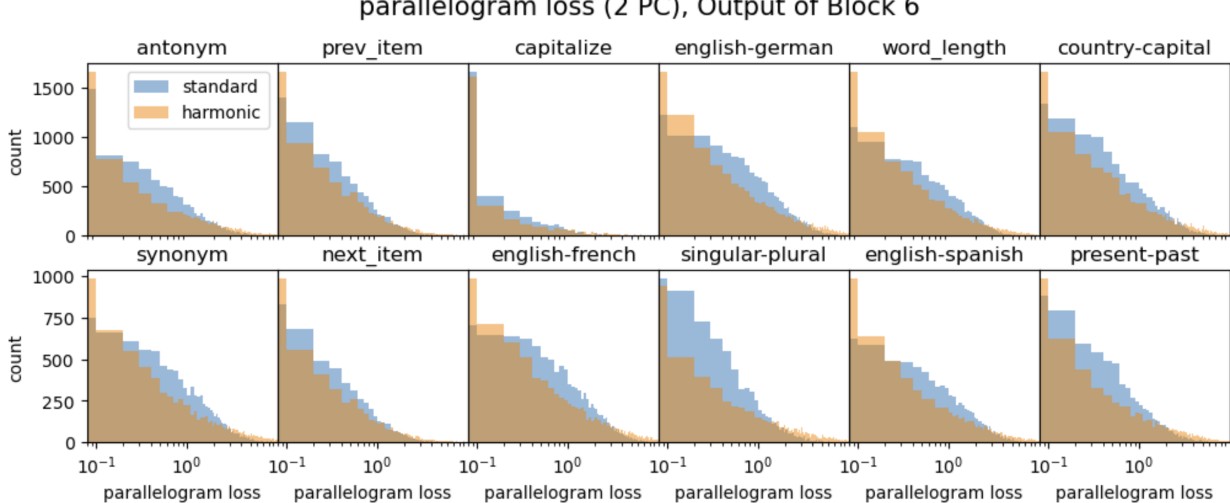

Figure 8: Harmonic loss (harmonic) and cross-entropy loss (standard) induce qualitatively different representations in the intermediate layer 6 of GPT2. We show the distribution of parallelogram loss for the parallelogram dataset. Harmonic loss has more perfect parallelograms (spike close to zero loss) but demonstrates a heavier tail.

## B  Identifying Coset Structure in Permutation Representations

To explore the coset structure in permutation representations of $S_4$, we began by enumerating its subgroups. Using this enumeration, we computed all possible left and right cosets of each subgroup in $S_4$, yielding 28 distinct left cosets and 28 distinct right cosets.

Among these cosets, two pairs are equivalent, since we consider two of the four normal subgroups of $S_4$: the alternating group $A_4$ and the Klein-4 group. To focus on meaningful structures, the trivial subgroup and the entire group were excluded from further analysis.

The coset partitions were then compared using the silhouette score, a metric for evaluating the quality of clustering. This comparison helped identify the partition with the most structured coset organization, which is likely the structure that the model has captured during training. We then color the representation according to the best-clustered partition, with each coset being a different color.

## C  Analyzing GPT2 hidden representations

In Section 6, we have shown that GPT2 trained with the harmonic loss has nicer structures in its embeddings (i.e., parallelograms) than that trained with the standard cross-entropy loss. We now show that intermediate representations (output of Block 6) induced by the harmonic loss are also qualitatively different from those of the cross-entropy loss. In Figure 8, the harmonic loss produces more perfect parallelograms (spike around zero parallelogram loss) but also displays a heavier tail for the parallelogram loss. The heavy tail is due to the heavy-tailedness of the harmonic loss (power law), as opposed to the cross-entropy loss (exponential). It remains to be understood if such heavy-tailedness is a feature or a bug for the harmonic loss, but the more perfect parallelograms are probably a good thing, or this at least suggests that imposing the harmonic loss at the end of the network can have noticeable influences in the intermediate representations. In Figure 9, we also notice that for the Captalize dataset, the lowercase and uppercase words tend to overlap in the first two PCs with the harmonic loss, but not with the cross-entropy loss. This again suggests the qualitative difference between the harmonic loss and the cross-entropy loss.

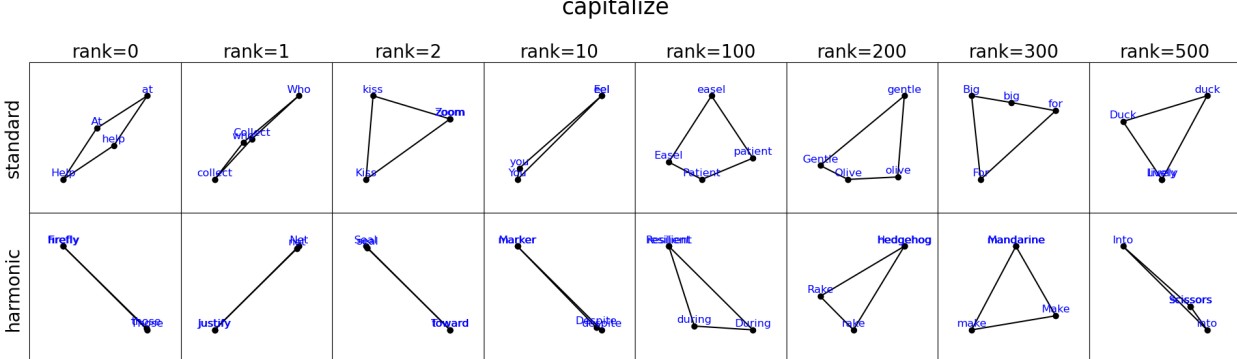

Figure 9: Visualization of layer 6 representations projected onto the first two principal components, for the capitalize dataset. The harmonic loss (bottom) tends to collapse corresponding lower-case and upper-case words, while the cross-entropy loss (top) places them at different locations.

## D    Comparison of Harmonic Loss to Alternative Loss Functions

We briefly contrast the harmonic layer ($\ell_2$ logits + HarMax) with three popular loss families. Throughout, let $\mathbf{x}$ be an example embedding, $\mathbf{w}_y$ the weight of the correct class $y$, and $\mathbf{w}_i$ those of incorrect classes.

**(a) Contrastive / InfoNCE.**  A generic form is

$$\mathcal{L}_{\text{contr}} = -\log \frac{\exp\big(s(\mathbf{x}, \mathbf{x}^+)/\tau\big)}{\exp\big(s(\mathbf{x}, \mathbf{x}^+)/\tau\big) + \sum_i \exp\big(s(\mathbf{x}, \mathbf{x}_i^-)/\tau\big)}.$$

It enforces only *relative* ordering $s(\mathbf{x}, \mathbf{x}^+) > s(\mathbf{x}, \mathbf{x}^-) + m$, so entire constellations can drift or rotate. In contrast, harmonic loss pulls every example directly toward a fixed class anchor $\mathbf{w}_y$ and repels it from all others, yielding a stable, globally referenced geometry.

**(b) Margin-based SoftMax.**  Large-margin variants add a fixed gap $\Delta$ to every class boundary, $s(\mathbf{x}, \mathbf{w}_y) \geq s(\mathbf{x}, \mathbf{w}_i) + \Delta$. Because $\Delta$ is global, semantically close classes (e.g. dog vs. cat) are forced as far apart as unrelated ones (dog vs. airplane). Harmonic loss adapts separation dynamically: $p_i \propto \|\mathbf{x} - \mathbf{w}_i\|^{-n}$, so related concepts converge while unrelated ones diverge, yielding meaningful hierarchies (e.g. the FAMILY-TREE task).

**(c) Spherical / cosine losses.**  These constrain embeddings to the unit hypersphere and optimise angular margins: $\mathcal{L}_{\text{sph}} = -\log \frac{e^{s \cos \theta_y}}{\sum_i e^{s \cos \theta_i}}$. While scale-invariant in angular space, they ignore absolute Euclidean proximity; our tasks (lattice, modular-add) benefit from the latter, explaining the poorer alignment of spherical loss.

We also run some experiments contrasting harmonic loss with loss **(a)** contrastive loss and **(c)** spherical loss for the in-context learning and modular addition tasks. Results for MLP and Transformer models are in Figure 10 and Figure 11, respectively.

## E    Sweeping HarMax Exponent Value

We perform experiments sweeping the HarMax exponent value for the in-context learning and modular addition tasks. Results are displayed in Figure 12 and Figure 13. We note that varying $n$ has minor impacts on lattice quality, with the default choice $n=1$ having the highest explained variances. Based on the modular addition task, our overall takeaway is that MLPs prefer the default $n=1$, while explained variance and circular structure for Transformer representations may improve with a slightly larger exponent.

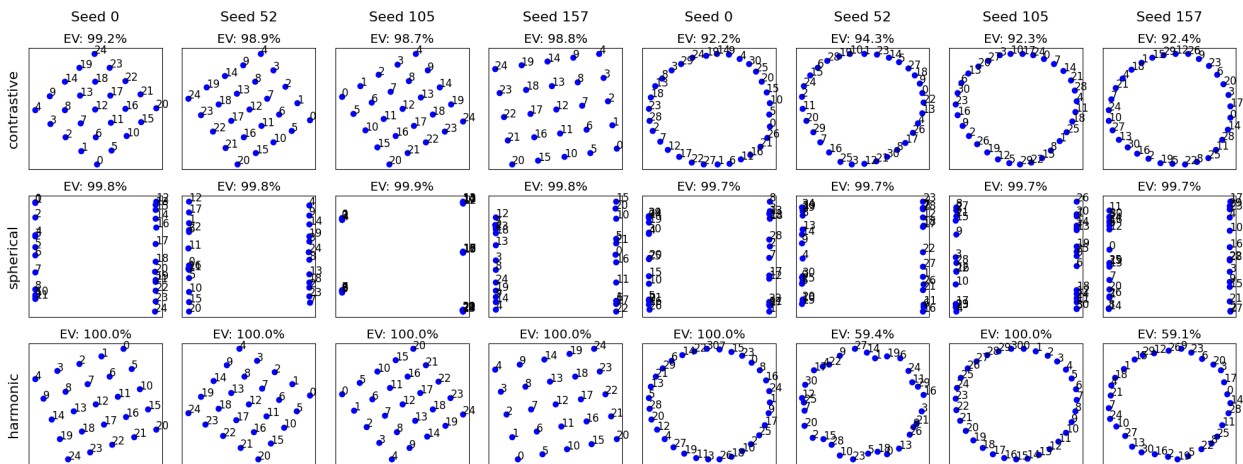

Figure 10: Results for MLP models. Rows show harmonic, DotProd+HarMax, $\ell_2$ +SoftMax, and standard losses (top to bottom). Harmonic loss achieves the best reconstruction across seeds.

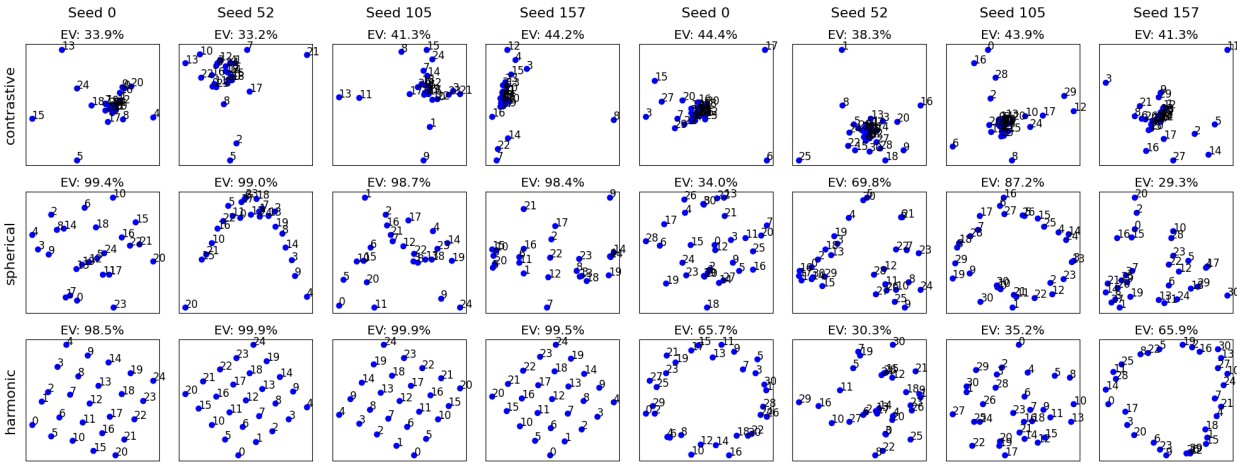

Figure 11: Results for Transformer models. Same ordering as Fig. 10.

# F   Properties of Harmonic Loss: Proofs

**Theorem 1** (**Finite Convergence of Harmonic Loss**). *Consider a classification model with $K$ classes and weight vectors $w_1, \ldots, w_K \in \mathbb{R}^d$ (no bias). Let $\{(x_i, y_i)\}_{i=1}^n$ be the training set, with $y_i \in \{1, \ldots, K\}$. The cross-entropy loss is given by*

$$L_{\mathrm{CE}}(W) = -\sum_{i=1}^n \ln \frac{\exp(w_{y_i} \cdot x_i)}{\sum_{j=1}^K \exp(w_j \cdot x_i)}.$$

*The harmonic loss (with exponent $\beta > 0$) is given by*

$$L_{\mathrm{H}}(W) = -\sum_{i=1}^n \ln \frac{\|x_i - w_{y_i}\|^{-\beta}}{\sum_{j=1}^K \|x_i - w_j\|^{-\beta}}.$$

*If the training data is linearly separable (i.e. there exists $W$ such that for all $i$, $w_{y_i} \cdot x_i > w_j \cdot x_i$ for $j \neq y_i$), then:*

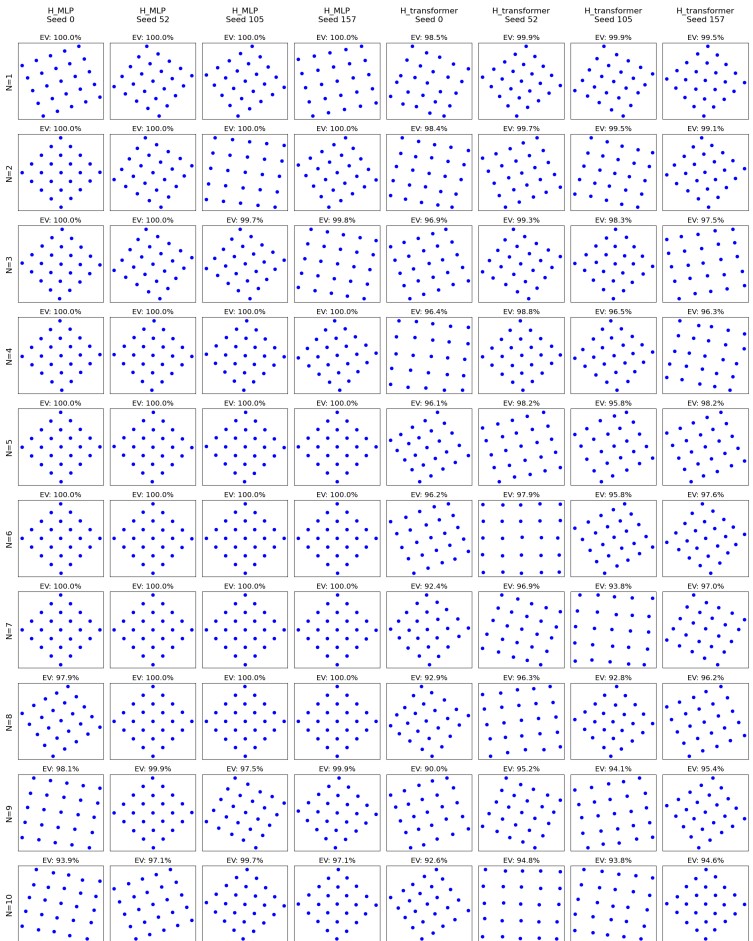

Figure 12: Effect of the harmonic exponent $n$ on lattice in-context learning. We sweep $n \in \{1, \ldots, 10\}$. Columns 1–4: Harmonic–MLP, columns 5–8: Harmonic Transformer. The learned $5 \times 5$ lattice is remarkably stable; $n$=1 already provides crisp and interpretable geometry.

- $L_{\mathrm{CE}}(W)$ *has no finite minimum. In fact, for any weight matrix $W$ that classifies all training points correctly, one can decrease $L_{\mathrm{CE}}$ further by scaling $W$ to larger norm. Thus the infimum of $L_{\mathrm{CE}}$ is $0$ but it is approached only as $\|W\| \to \infty$.*

- $L_{\mathrm{H}}(W)$ *attains a (global) minimum at some finite $W$. Once the weights are large enough to classify all training points correctly (i.e. $\|x_i - w_{y_i}\| < \min_{j \neq y_i} \|x_i - w_j\|$ for all $i$), increasing the norm of $W$ does not reduce $L_{\mathrm{H}}$. In particular, $L_{\mathrm{H}}$ is* scale-invariant*: scaling all $w_k$ and all $x_i$ by a common factor leaves the loss unchanged. Consequently, $L_{\mathrm{H}}$ has a finite global minimizer.*

*Proof.* For the cross-entropy loss $L_{\mathrm{CE}}$, suppose $W$ classifies all training examples correctly. Then for each $i$, $w_{y_i} \cdot x_i > \max_{j \neq y_i} w_j \cdot x_i$. Consider scaling $W$ by a factor $t > 1$: replace each $w_k$ with $tw_k$. Then $w_{y_i} \cdot x_i$ and $w_j \cdot x_i$ are both multiplied by $t$. The SoftMax probability of the true class $y_i$ becomes

$$P_W(y_i|x_i) = \frac{\exp(w_{y_i} \cdot x_i)}{\sum_j \exp(w_j \cdot x_i)}.$$

Under scaling $tW$, this becomes

$$P_{tW}(y_i|x_i) = \frac{\exp(t\, w_{y_i} \cdot x_i)}{\sum_j \exp(t\, w_j \cdot x_i)}.$$

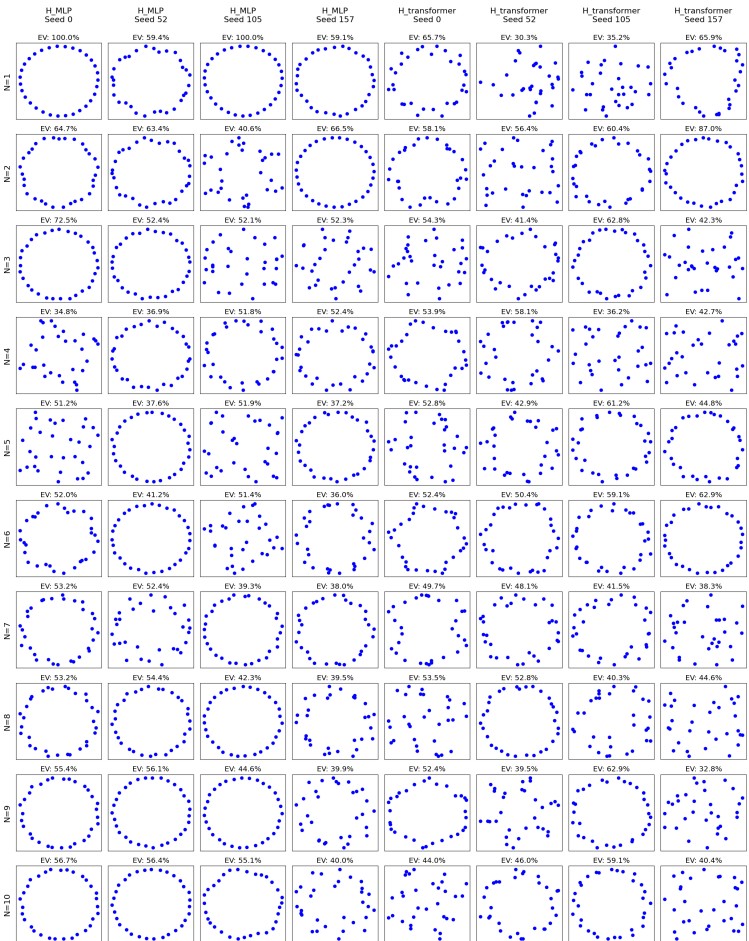

Figure 13: Effect of the harmonic exponent $n$ on modular addition. Columns 1–4: Harmonic–MLP, columns 5–8: Harmonic Transformer. MLPs remain stable across seeds, whereas Transformers are more sensitive yet form tighter circles at higher $n$; $n=1$ works well for MLPs, while a larger $n$ may benefit Transformers.

Since $w_{y_i} \cdot x_i$ is the largest logit for sample $i$, as $t \to \infty$ we have $P_{tW}(y_i|x_i) \to 1$ and thus $-\ln P_{tW}(y_i|x_i) \to 0$. This holds for all $i$, so $L_{\mathrm{CE}}(tW) \to 0$ as $t \to \infty$. Therefore, no finite $W$ minimizes $L_{\mathrm{CE}}$; the infimum 0 is approached only in the limit $\|W\| \to \infty$.

For $L_{\mathrm{H}}$, once $W$ is such that each training point is correctly classified by its nearest prototype (i.e. $\|x_i - w_{y_i}\| < \|x_i - w_j\|$ for all $j \neq y_i$), increasing the norms $\|w_k\|$ further will not improve the loss. In fact, if every $x_i$ is closer to its correct $w_{y_i}$ than to any other $w_j$, then the harmonic probabilities

$$P_W(y_i|x_i) = \frac{\|x_i - w_{y_i}\|^{-\beta}}{\sum_{j=1}^{K} \|x_i - w_j\|^{-\beta}}$$

remain unchanged under a uniform scaling: if we replace $x_i$ by $cx_i$ and $w_k$ by $cw_k$, then $\|cx_i - cw_k\| = c\|x_i - w_k\|$, so the scaling factors cancel. Therefore, once correct classification is achieved, no further reduction in loss is obtained by increasing $\|W\|$, and $L_{\mathrm{H}}$ achieves its minimum at finite $W$. $\qquad \square$

**Theorem 2** (**PAC-Bayesian Generalization Bound of Harmonic Loss**). *Assume all training examples lie within a ball of radius $R$ in input space, i.e. $\|x_i\| \leq R$ for all $i$. Suppose a weight matrix $W$ achieves a distance margin of $\gamma > 0$ on the training set, meaning that for every training sample $(x_i, y_i)$ and any other class $j \neq y_i$,*

$$\|x_i - w_{y_i}\| + \gamma \leq \|x_i - w_j\|.$$

*Then, with probability at least $1 - \delta$, the generalization (test) error of the harmonic classifier satisfies*

$$\Pr_{(x,y)\sim D} \left[ h_W(x) \neq y \right] \;\leq\; \mathcal{O}\left( \frac{R\,\|W\|}{\gamma\sqrt{n}} + \sqrt{\frac{\ln(1/\delta)}{n}} \right),$$

*where $h_W(x)$ denotes the predicted class and $n$ is the number of training samples.*

*In particular, $\|W\|$ is finite for harmonic loss (by Theorem 1), and typically much smaller than the weight norm of the solution obtained with cross-entropy loss. Thus, the harmonic classifier has a tighter generalization bound.*

*Proof.* Applying the standard PAC-Bayes margin bounds (see e.g. Neyshabur et al. (2017)), one obtains that with probability at least $1 - \delta$,

$$\Pr(h_W(x) \neq y) \leq \mathcal{O}\left( \frac{R\,\|W\|}{\gamma\sqrt{n}} + \sqrt{\frac{\ln(1/\delta)}{n}} \right).$$

Since the harmonic loss yields a solution with finite $\|W\|$, the bound is finite. In contrast, the cross-entropy solution would have $\|W\| \to \infty$ even when achieving zero training error, rendering a similar bound meaningless. $\qquad\square$

**Theorem 3** (**Interpretable Representations of Harmonic Loss**). *At a critical point (in particular, a global minimum) of the harmonic loss, each weight vector $w_k$ becomes an interpretable class center for class $k$. Specifically, the stationarity condition implies*

$$w_k \;=\; \sum_{i:y_i=k} \alpha_i\, x_i \qquad \text{with } \alpha_i \geq 0, \; \sum_{i:y_i=k} \alpha_i = 1,$$

*i.e. $w_k$ is a convex combination of the training examples of class $k$. Consequently, $w_k$ represents the center point of its class, leading to more interpretable representations compared to cross-entropy loss.*

*Proof.* Differentiate the harmonic loss with respect to $w_k$. For simplicity, denote

$$p_i^k \;=\; \frac{\|x_i - w_k\|^{-\beta}}{\sum_{j=1}^{K} \|x_i - w_j\|^{-\beta}}.$$

For samples $x_i$ with $y_i = k$, the derivative takes the form

$$\frac{\partial L_{\mathrm{H}}}{\partial w_k} \;=\; -\sum_{i:y_i=k} \frac{\beta}{\|x_i - w_k\|^2}\,(w_k - x_i)\, p_i^k + \text{terms from } i \text{ with } y_i \neq k.$$

At a critical point, the total derivative vanishes. Rearranging the stationarity conditions (and noting that the repulsive forces from other classes tend to balance out overall on average due to long distance) yields

$$w_k = \frac{\sum_{i:y_i=k} \frac{1}{\|x_i-w_k\|^2} x_i + \sum_{j:y_j\neq k} \frac{1}{\|x_j-w_k\|^2} x_j}{\sum_{i:y_i=k} \frac{1}{\|x_i-w_k\|^2} + \sum_{j:y_j\neq k} \frac{1}{\|x_j-w_k\|^2}}.$$

Since $w_k$ is closer to class-$k$ examples than to others, the weights $\frac{1}{\|x_i-w_k\|^2}$ for $i$ with $y_i = k$ dominate the sum. Define

$$\alpha_i = \frac{\frac{1}{\|x_i-w_k\|^2}}{\sum_{i:y_i=k} \frac{1}{\|x_i-w_k\|^2} + \sum_{j:y_j\neq k} \frac{1}{\|x_j-w_k\|^2}}.$$

Then $w_k$ can be written as a convex combination

$$w_k = \sum_{i:y_i=k} \alpha_i\, x_i + \sum_{j:y_j\neq k} \alpha_j\, x_j.$$

Table 1: Validation accuracy on ImageNet using different loss functions.

| Loss | Top-1 Val Acc | Top-5 Val Acc |
|---|---|---|
| Cross-Entropy (Our Impl.) | 74.17% | 91.88% |
| Harmonic Loss (Our Impl.) | 75.08% | 92.12% |
| Cross-Entropy (Khosla et al., 2020) | 77.6% | 95.3% |
| Supervised Contrastive Loss (Khosla et al., 2020) | 78.7% | 94.3% |

Table 2: Probing F1 score on SST-2 and CoLA datasets.

| Model | SST2 (Layer 0) | CoLA (Layer 0) | SST2 (Layer 6) | CoLA (Layer 6) |
|---|---|---|---|---|
| Cross-Entropy | $76.2 \pm 1.5\%$ | $73.9 \pm 1.0\%$ | $79.9 \pm 1.3\%$ | $78.2 \pm 1.7\%$ |
| Harmonic | $77.9 \pm 1.1\%$ | $74.3 \pm 1.0\%$ | $79.9 \pm 1.7\%$ | $77.1 \pm 4.2\%$ |

In many practical settings, the contribution from $x_j$ with $y_j \neq k$ is negligible, so $w_k$ is nearly a convex combination solely of class-$k$ samples. By construction, $\alpha_i \geq 0$ and the weights sum to 1. This shows that $w_k$ is an interpretable vector representing its class center. In contrast, for cross-entropy loss the stationary condition does not yield a similar expression for $w_k$ as a combination of data points. □

**Remark**: Under cross-entropy loss, the weight vectors usually end up pointing to the average direction of class elements, due to its use of the dot product. However, they do not have a closed-form formula like the harmonic loss above, and the weight vectors are not *linear* combinations of all class feature directions. We believe that enforcing such linear combination structure plays a crucial role in enhancing interpretability – it directly aligns with the Linear Representation Hypothesis (Park et al., 2023), and natively supports compositional generalization.

# G  Additional Benchmark Results

## G.1  ImageNet

ImageNet (Deng et al., 2009) is a large-scale visual dataset commonly used in object recognition research. We compare the performance of standard cross entropy loss and harmonic loss on ImageNet. We trained ResNet-50 with AutoAugment data augmentation method for 90 epochs, starting with a learning rate of 0.1, which was reduced by a factor of 10 at epochs 10, 30, 60, and 80. The training results are presented in Table 1. We have also implemented our own cross-entropy training pipeline, and compared them with existing results in (Khosla et al., 2020). In our implementation, the harmonic model modestly outperformed the standard model.

## G.2  SST2 and GLUE

We also compare the standard GPT2 and harmonic GPT2 with the GLUE benchmark below. We evaluate two tasks, COLA (linguistic acceptability) (Warstadt et al., 2018) and SST2 (sentence sentiment classification) (Socher et al., 2013). We train a 1-layer MLP probe with hidden dimension 16 that takes the model's residual stream representation as an input, and outputs the label. Table 2 shows the F1 score of the probe on validation dataset.

