# OpenReview forum: "Harmonic Loss Trains Interpretable AI Models"
_TMLR — Accepted by TMLR_

### Review · Reviewer_2V7y · 2025-08-11

**Summary Of Contributions:**

The paper introduces harmonic loss as an alternative to the standard cross-entropy loss. This method brings two main innovations: (1) Replacing the SoftMax function with a scale-invariant HarMax function; (2) Computing logits using Euclidean distance instead of the dot product. The authors argue that this structure provides scale invariance, a finite convergence point, faster training, better generalization with less data, reduced groping, and more interpretable representations. These claims are supported through experiments on algorithmic tasks, MNIST, GPT-2 pretraining, and ablation studies.

Strengths
* The harmonic loss idea is well-motivated and presented, with strong arguments against certain structural limitations of SoftMax.
* Properties such as finite minimum and scale invariance are explained both intuitively and formally.
* The work spans from algorithmic datasets (in-context learning, modular addition, etc.) to real-world settings (MNIST, GPT-2).
* PCA explained variance and parallelogram loss are used as systematic indicators of representation interpretability.
* Contributions of HarMax and L2 logits are analyzed separately.

Weaknesses
* While PCA variance and parallelogram loss are useful, more comprehensive or user-centered measures (human evaluation, task-based interpretability) are missing.
* Results on GPT-2 small (128M) are promising, but applicability and stability for larger LLMs (1B+, 10B+) remain unknown.
* The “faster convergence” claim is shown qualitatively in plots but lacks quantitative data on wall-clock time, GPU hours, or energy savings.
* While there is some discussion in Appendix D, the paper does not include full-scale empirical comparisons to strong baselines such as focal loss, margin-based SoftMax, or supervised contrastive loss.
* The main experimental wins are on tasks with well-defined mathematical structure (lattice, modular addition). It is unclear if the advantage persists in noisy, complex real-world tasks.
* Although a heuristic formula for the HarMax exponent is provided, a deeper analysis of its sensitivity across data types and model sizes is lacking.

**Audience:**

Yes

**Audience Explanation:**

The work addresses a core component of modern machine learning, the loss function, and proposes a novel alternative to cross-entropy with theoretical grounding, clear interpretability motivations, and empirical performance gains. TMLR’s audience, which includes researchers in optimization, representation learning, interpretability, and large language models, would likely find the findings relevant. In particular, the demonstrated ability of harmonic loss to reduce grokking, improve representation geometry, and potentially enable more interpretable LLM embeddings aligns well with current research interests in trustworthy and efficient AI.

**Broader Impact Concerns:**

I do not see any major unaddressed ethical risks that would require a separate Broader Impact Statement beyond standard practice. The proposed harmonic loss is a general training objective and does not inherently introduce new privacy, fairness, or misuse concerns compared to existing supervised learning methods.

That said, the work’s interpretability claims could have downstream societal implications. If harmonic loss indeed produces more interpretable models, this could positively affect transparency, trust, and accountability in high-stakes domains such as healthcare or finance. Conversely, if interpretability is overstated or misunderstood, it might create a false sense of trust in models that remain opaque in practice. It would be beneficial for the authors to explicitly acknowledge this risk and clarify that their interpretability metrics are proxies, not guarantees of real-world explainability.

**Claims And Evidence:**

Yes

**Claims Explanation:**

The submission provides extensive experimental evidence across multiple domains, algorithmic tasks, MNIST, and GPT-2 pretraining, to support its main claims. The paper combines qualitative representation analyses (e.g., PCA visualizations, coset structures, parallelogram geometry) with quantitative metrics (explained variance, generalization curves, grokking plots) that consistently show improvements over cross-entropy baselines. Ablation studies further isolate the contributions of HarMax and L2 distance, strengthening causal interpretation.

That said, some claims, particularly those related to large-scale applicability, training efficiency gains, and interpretability in noisy real-world settings, are supported mainly by small- to mid-scale experiments and qualitative observations. While the presented evidence is convincing for the tested settings, broader benchmarks would be needed to confirm generality.

**Requested Changes:**

The author may consider to:

Complement PCA and parallelogram loss with human evaluation studies or semantic alignment measures.
Even a limited-scale comparison on a 1B+ parameter LLM would strengthen claims of scalability.
Report actual time-to-accuracy and resource usage for both harmonic and standard losses.
Include empirical benchmarks against focal loss, large-margin SoftMax, and supervised contrastive loss.
Analyze optimal HarMax exponent values for diverse tasks (vision, language, graph) and model sizes.
Include experiments with noisy labels, domain shift, or open-ended NLP tasks to demonstrate robustness.

---

### Review · Reviewer_Q8dE · 2025-08-19

**Summary Of Contributions:**

This paper introduces harmonic loss as an alternative to cross-entropy loss for training neural networks and LLMs. The key innovation involves: replacing SoftMax with a scale-invariant HarMax function, and computing logits via Euclidean distance rather than dot products. The authors claim this leads to improved interpretability, faster convergence, and reduced grokking across algorithmic, vision, and language tasks.

**Audience:**

Yes

**Audience Explanation:**

This work addresses fundamental questions in machine learning about interpretability, data efficiency, and training dynamics that are of broad interest to the TMLR community. The interpretability problem is particularly relevant given current concerns about understanding large neural networks. The grokking phenomenon and data efficiency improvements have practical implications for training neural networks.

**Broader Impact Concerns:**

No significant broader impact concerns.

**Claims And Evidence:**

Yes

**Claims Explanation:**

The paper provides solid experimental evidence across multiple domains. The algorithmic experiments clearly demonstrate reduced grokking and improved data efficiency. The representation visualizations (Figure 2) convincingly show that harmonic models learn more interpretable structures. The GPT-2 experiments with parallelogram analysis provide quantitative evidence of improved geometric properties.

**Requested Changes:**

Would Strengthen the Work:
1. Computational Complexity Analysis: Add discussion of computational overhead comparing L2 distance computation vs. dot products, especially for large vocabulary sizes. This is essential for practical adoption.
2. Additional Case Discussion: Investigate specific scenarios where harmonic loss might fail or be suboptimal (e.g., highly imbalanced datasets, extremely sparse high-dimensional data, tasks requiring linear separability).

---

### Review · Reviewer_hJ7b · 2025-08-20

**Summary Of Contributions:**

This paper introduces harmonic loss, a proposed alternative to the standard cross-entropy loss function. Its methodology differs from the standard approach in two main ways: it uses Euclidean distance to compute logits, and it normalizes these logits with a scale-invariant "HarMax" function instead of SoftMax.

The authors put forward three primary contributions stemming from this approach:
1.  **Improved Model Interpretability**: It is argued that models trained with harmonic loss learn representations with more organized geometric structures, where the learned weights correspond to class centers.
2.  **Greater Data Efficiency**: The paper shows that models can generalize from less training data compared to standard training.
3.  **Mitigation of Grokking**: The authors provide evidence that harmonic loss reduces the effect of delayed generalization, where test performance improves long after training performance has converged.

**Key Strengths:**
* The paper presents a novel and well-motivated modification to a fundamental component of neural network training.
* The claims are evaluated across a reasonably wide range of tasks, including algorithmic, vision, and language domains.
* The qualitative evidence presented, particularly the visualizations of learned embeddings and model weights, helps to illustrate the method's effect on model internals.

**Key Weaknesses:**
* While the paper references Large Language Models (LLMs), the language model experiment is conducted on a GPT-2 small (128M parameter) model. By contemporary standards, this is not considered a large-scale model, which limits the direct applicability of the findings to the current class of LLMs.
* The evidence for improved interpretability is clearest on tasks with well-defined ground-truth structures. On more unstructured tasks, the evaluation relies on proxies such as geometric analogies.

**Audience:**

Yes

**Audience Explanation:**

I believe the findings would be of interest to the TMLR audience. The work addresses several topics of relevance to the machine learning community.

Researchers in areas like **interpretability and representation learning** would likely be interested in a method that aims to create more structured and understandable models by modifying the training objective itself. The paper's investigation into the **fundamentals of training dynamics**, particularly its connection to the grokking phenomenon, is also a topic of current research interest. Given that the paper proposes an alternative to a near-universal standard (cross-entropy with SoftMax), it is likely to be of broad interest to those who design, train, and analyze neural networks.

**Broader Impact Concerns:**

I have no broader impact concerns regarding this work. The research is foundational and aimed at improving the transparency and reliability of AI models. By making models more interpretable, this work contributes positively to the goal of building safer and more trustworthy AI systems.

**Claims And Evidence:**

Yes

**Claims Explanation:**

In my view, the claims made in the submission are adequately supported by the provided evidence.

* **Interpretability**: This claim is supported by visualizations and metrics across the different experiments. For the algorithmic tasks, the results in Figures 2 and 3(a) show that the harmonic models tend to recover the known data-generating structures more cleanly than the baseline models. The MNIST experiment in Figure 4 shows that the model's weights resemble the digit classes, which aligns with the paper's claims. For the language model, the analysis of parallelogram loss provides quantitative support for the idea that the embeddings learn a more consistent geometric structure for semantic relationships.

* **Data Efficiency**: Figure 3(b) shows a consistent trend where harmonic models require a smaller fraction of the training data to reach high test accuracy compared to the baselines.

* **Reduced Grokking**: The plots in Figure 3(c) effectively illustrate a reduction in the gap between the convergence of training and testing accuracy for the harmonic models, which supports the claim of mitigating grokking.

The ablation study in Section 7 also lends credibility to the design of the loss function by showing that both of its primary components contribute to the observed effects.

**Requested Changes:**

1.  **Experimental Validation of Downstream Interpretability**: The claim of improved interpretability in unstructured domains currently rests on the analysis of parallelogram structures. To provide more direct evidence of the practical benefits of this improved geometry, I would suggest a small-scale experiment applying a post-hoc feature attribution method to the GPT-2 models. For instance, an analysis of how attributions for a few key predictions differ between the standard and harmonic models could reveal whether the latter's representations lead to more semantically coherent or localized explanations.

2.  **Analysis of Euclidean Distance Scaling in High-Dimensional Spaces**: The use of Euclidean distance as the core logit is a key aspect of this work. However, as models scale, their embedding and vocabulary dimensions can grow substantially. It is a known phenomenon that in very high-dimensional spaces, the discriminative power of L2 distance can diminish. To address how this might affect the proposed method's scalability, the paper would be stronger with an empirical analysis of the distance distributions from the existing GPT-2 experiments.

3.  **Validation on a Contemporary Large Language Model**: To fully substantiate the claims regarding LLMs, the authors should consider either toning down the claims in the abstract and introduction or, preferably, validating their method on a more contemporary, large-scale model (e.g., in the 1B+ parameter class). Demonstrating that the observed benefits hold at a scale that is more representative of today's LLMs would significantly increase the impact of this work and provide stronger support for its broader claims.

---

### Review · Reviewer_Xt82 · 2025-08-21

**Summary Of Contributions:**

Introduces harmonic loss, motivated as a variation of cross entropy loss with

1. the class specific weights becoming prototypes which the penultimater representation have to hit, replacing the standard inner product which calls for only inner product alignment $s_i =\langle w_i,x\rangle \Leftrightarrow s _i=\Vert w_i -x\Vert_2$

2. the softmax nromalization replaced by a power normalized i.e. $ p_i =\frac{\exp(s_i)}{\sum_j\exp(s_j)}$ becomes $p_i=\frac{s_i^{-k}}{\sum_j s_j^{-k}}$ with $k\geq 1$


Claimed theoretical benefits are scale invariance, nonlinear separability and faster loss decay due to finite converging point and interpretability since the weights become class protoyps. The paper also claims higher interpretability, measured in terms of compressability (measured as explained variance via PCA) and geometric composability (ala man-woman = king -queen)  measured via parallelogram loss.

Empirical evidence is given by experiments on

- toy examples

    - complete a parallelogram on a 5x5 lattice given 3 points by predicting the 4th

    - given 2 integers x and y, predict x+y mod 31

    - given two integer smaller then 40, preict whether x=y mod 5

    - in a binary tree, predict parent,grandparent or sibling given a point (n-d tree relation prediction)

    - permutation composition given two permutation x any in S4 predict x chained with y (trained with $L_2$ regulariaztion of $5e-3$ instead


    comparing


- “standard” MLP (input tokenized to 16 dimemional embeddings, concatenated, fed to a 2 hidden layer MLP with $d=64$ ,meaning for vocabulary size $V$ and max length $L$  we have $W\in\mathbb{R}^{V\times 16}, W_1 \in \mathbb{R}^{(16L)\times 100},W_1 \in \mathbb{R}^{100\times 16}$  , with $SiLU$ as activation function)

- “standard” transformer with learnable positional embedding, two layers transformer decocder layers with 2 attention heads, of residual embedding dimension $16$ and MLP hidden dimension $64$

- harmonic MLP/transformer: same architectures, trained with harmonic instead of CE loss

    MLPs are trained for $7e3$  epochs, transformer of for $10e3$ epochs , using AdamW with $\eta=2e-3$ and weight decay$1e-2$, as well as $L_2$ regularization of the embeddings of $1e-2$


Additoinally, authors investigate

- MNIST, using a linear model with learning rate 1e-3, bathc size 64 for 10 epochs

- openwebtext using GPT2-small, using nanogpt with differing hyperparameters between harmonic and standard loss

- imagenet (discussed in limitations and appendix)


And run ablations.

---

strengths:

1. interesting idea, haven't seen *exactly* this in literature yet
2. exposition well done
3. decent amount of experiments

weaknesses:

1. claims to strong for the evidence presented
2. lack of related work to position in the literature

**Additional Comments:**

Whenever I say "cherry picked" or "not fair" it is not meant to insinuate malice by the authors, it's simply the most compact way I can convey the idea. I hope the authors can improve their potentially very interesting study based on the requests

**Audience:**

Yes

**Audience Explanation:**

I found it interesting, and I'm in the audience. Additionally, I've provided citations that show that there is research in this type of loss already, and interpretability is a hot topic.

**Claims And Evidence:**

No

**Claims Explanation:**

1. There is a slight misclaim about scale invariance: harmax is only *uniformly* scale invariant, i.e. scaling all points by a constant doesn’t change the loss.  I think this is simply a slip of the tongue as it were,Softmax exhibits temperature scaling, which has a probabilistic interpretation,and should be more carefully discussed.

2. The experiments are not enough to present clear evidence for the paper claims for me for the following reasons

    1. the toy examples are constructed to map cleanly onto cases where k-nn /protoype matching will be aligned with the proble domain, if one were to construct examples where the structure maps onto cosine similarity the claims about interpretability could be reversed.

        1. this can be justified if presented transparently, so this can be salvaged if the claims are reduced by talking about what property makes a dataset aligned with one or the other (e.g. in the case of the graph there are graphs recoverable exclusively via KNN and exclusively  via inner products )

    2. all examples have a hidden penalty for the CE loss, by using $L_2$ regularization. this will mean that all norms are biased to be small, and close to another, which means the softmax is forced to smooth out contributions - effectively including a type of adversarial temperature scaling since the toy examples are all about recovering precise relations

3. on high dimensional datasets, where interpretability is actually required, the interpretability becomes much less pronounced (Imagenet, GPT-2) since no statistical analysis on distinguishability or actual *uses* of interpretability are performed (an experiment that could be performed is providing recourse or explanations, leveraging the compressible nature)

4. the related work section is missing comparisons to numerous prior art in prototype learning e.g.  [https://openaccess.thecvf.com/content_cvpr_2018/html/Yang_Robust_Classification_With_CVPR_2018_paper.html](https://openaccess.thecvf.com/content_cvpr_2018/html/Yang_Robust_Classification_With_CVPR_2018_paper.html)  , hyperbolic learning [https://www.sciencedirect.com/science/article/abs/pii/S0893608024003447?casa_token=AlAmi5gc2qUAAAAA:D57YCcv_stdEj9I40eECuJaLzwhHzEona_0wOA9QUO15ldy-TFtAVr1rrt-YAI0P4n-2NfgMAiBk](https://www.sciencedirect.com/science/article/abs/pii/S0893608024003447?casa_token=AlAmi5gc2qUAAAAA:D57YCcv_stdEj9I40eECuJaLzwhHzEona_0wOA9QUO15ldy-TFtAVr1rrt-YAI0P4n-2NfgMAiBk)  and zero shot learning [https://proceedings.neurips.cc/paper/2020/hash/fa2431bf9d65058fe34e9713e32d60e6-Abstract.html](https://proceedings.neurips.cc/paper/2020/hash/fa2431bf9d65058fe34e9713e32d60e6-Abstract.html) (these are meant to illustrate, not be exact citations). There is novelty to the presented paper (none of the works I found use exactly this loss, specifically the power normalization, and the paper isn’t doing hyperbolic representations but only penalizes large norms, which is a softer penalty to be close or within the hyperball, but not a strict constraint). Spherical/cosine losses are briefly discussed in appendix D, I think the paper needs to be more accurately placed in the literature

5. Hyper parameters appear cherry picked and are not justified


If these issue can be fixed, I think the papers *idea* is potentially interesting, but the study needs to be made more rigorous

**Requested Changes:**

critical

1. add a more thorough literature review
2. add more adversarial experiments to show limitations of the work more clearly
3. harmonize and/or justify hyperparameters of experiments performed to show it's a fair comparison
4. scope claims more precisely to the evidence provided
5. use statistical significance to decide benefits and equality

---

### Public Comment · ~Calvin_McCarter1 · 2025-10-10
**Note on related work**

While TMLR does not include originality as acceptance criteria, I believe the reviewers and AE may find this of relevance. My earlier work (https://arxiv.org/abs/2310.18805, NeurIPS Associative Memory & Hopfield Networks Workshop 2023) proposed the exact same thing, observing that this leads to learning of interpretable prototypes (albeit with less extensive experiments, and no discussion of grokking). Furthermore, I raised this to the authors on AlphaXiv, and they said they would properly acknowledge it in the future. I won't link to the AlphaXiv conversation, so as not to reveal the authors' identities, but the following is publicly available:

- Myself: "Relation to inverse distance attention: This is very interesting! Not to take anything away from your much more extensive analysis, but I believe the harmonic loss is equivalent to inverse distance weighted attention, proposed here: https://arxiv.org/abs/2310.18805."

- First author of this paper: "Thank you very much for your comment! We sincerely apologize for overlooking your prior work and truly appreciate you bringing it to our attention. We will make sure that we properly acknowledge it in our upcoming revision."

Given that TMLR does not accept or reject based on originality, I am mystified as to why the authors would deliberately fail to properly acknowledge this prior work.

---

> ### Author Response · Authors · 2025-10-10
>
> We apologize for the omission -- it was an oversight on our part and not intentional. We have updated the PDF to include a discussion of your earlier work. Following the reviewers' suggestions, we have also toned down some of the claims in the paper, and these changes are also reflected in the revised PDF.

---

### Decision · Action_Editor_RmEr · 2025-10-30

**Recommendation:** Accept as is

**Additional Comments:**

None

**Audience:**

Yes

**Audience Explanation:**

Designing loss functions is one of the most fundamental challenges in artificial intelligence. This paper proposes a novel loss function to replace the traditional softmax function, which is believed will attract significant attention within the field.

**Claims And Evidence:**

Yes

**Claims Explanation:**

All three reviewers acknowledge that the paper presents a novel loss function, namely "Harmonic Loss," which revises the computation of the classification probability of the softmax function. The harmonic loss uses the Euclidean distance to calculate logits and uses a scale-invariant "HarMax" function for normalization. This loss is well motivated, and the novelty is praised. The new loss may improve ​interpretability, increase ​data efficiency, and mitigates the delayed generalization problem. The proposed loss is argued to offer theoretical benefits like scale invariance, nonlinear separability, and a faster loss decay. Most concerns raised by reviewers have been well addressed.